# Experimental investigation of bio-improved laterite for use as a subgrade material in road construction

**Reuben Orumah Sani[1]\*, Tobi Vincent Ogunro[2], Isaac Ibukun Akinwumi[1]**

1 Department of Civil Engineering, Covenant University, Ota, Ogun State, Nigeria, 2 Civil & Environmental Engineering, University of North Carolina, Charlotte, North Carolina, United States of America

☯ These authors contributed equally to this work.
\* reuben.sani@covenantuniversity.edu.ng

## Abstract

This study investigated the application of microbial-induced calcite precipitation (MICP) using the bacterium *Bacillus Anthracis* for laterite subgrade soil improvement in road construction. The California bearing ratio (CBR), unconfined compressive strength (UCS), and hydraulic conductivity test were conducted on the bacterium-modified soil. A quasi-field comparative test was also carried out to assess the strength properties of the bio-cemented and natural laterite. According to the American Association of State Highway Officials, the natural soil was classified as A–6 (5), a fair to poor soil for use as a subgrade material, justifying the need for improvement. The natural soil's CBR was improved by 80% at the $3 \times 10^8$ cells ml$^{-1}$ treatment. The soil was improved from "very soft to soft" to "stiff to very stiff" soil for all *Bacillus Anthracis* treatments, and the natural soils' UCS value increased by 191% at $3.0 \times 10^8$ cells ml$^{-1}$ *Bacillus Anthracis* treatment, confirming the CBR results. Best improvements at $3 \times 10^8$ cells ml$^{-1}$ of *Bacillus Anthracis*-treated soil provided better subgrade protection from water infiltration. Optimal bacterium-treated soil performed better in the field than in the laboratory. Results obtained successfully demonstrated the application of bio-improved laterite as a subgrade material for road construction.

## Introduction

Laterite soil is a common subgrade material used worldwide for road construction [1,2]. Despite being commonly used in road construction, laterite soils occasionally need stabilisation since they do not always meet predetermined standards [3]. According to Paulo et al. [4], most laterite soil improvement procedures, such as cement and lime improvement, have a negative environmental impact. Road construction projects must prioritise sustainable means of improving soil quality since there is a growing worldwide awareness of the need to reduce greenhouse gas (GHG) emissions and air pollution associated with the

**Data availability statement:** All relevant data are within the manuscript and its Supporting information files.

**Funding:** The author(s) received no specific funding for this work.

**Competing interests:** The authors have declared that no competing interests exist.

production of cement [5–7]. In recent decades, the number of cement substitutes for soil improvement, such as bio-cementation, has increased because of the global effort to achieve net-zero carbon dioxide emissions [8]. Microbial-induced calcite precipitation, MICP, is a method of improving soil properties using micro-organisms and cementation reagents. This method is more environmentally friendly [9,10]. Compared to the traditional cement and other materials [11–13] used for the soil improvement process, the MICP process produces minimal carbon dioxide ($CO_2$) and operates at room temperature [14]. Since MICP's discovery, the amount of soil improvement employing it for geotechnical goals has dramatically expanded [15].

Applications of MICP have been demonstrated in the following domains: environmental applications, where metals are stabilised [16]; development of biological shields for zonal remediation [17]; control of soil wind erosion [18–21]; Cementation of porous media and hydraulic control [22–26]. Researchers have also seen excellent results when employing bacteria to fix cracks in concrete structures [27–30]. Biogrouting of rock joints [31–36] and mitigating liquefaction risk using MICP have also yielded significant results [37,38].

Numerous studies have shown that MICP can improve the strength properties of granular soils. However, few field and laboratory investigations have investigated its applicability in road construction [39]. Chu and Wen [40] conducted the earliest study on the suitability of MICP for road construction. They created bio-cemented products to repair roads with good, unconfined compressive strength (UCS). Porter et al. [41] conducted a fruitful laboratory-scale trial to assess the efficacy of MICP in improving road bases. Their unconfined compressive strength was enhanced by 70%, yielding values appropriate for better road-based materials. Regretfully, sand is frequently utilised in study projects that are currently accessible, and there is little research on the application of MICP to enhance laterite soil for road construction. The possible reason for the scarcity of research works on laterite soil may be due to the understanding by researchers that the optimum grain size range of MICP-treated soils should be between 50 and 400 μm (i.e., approximately the soil particle sizes greater than sieve number 200, i.e., 75 μm aperture size) because it is difficult for bacterial activity to occur in very fine soils [42]. A review of the typical range of particle sizes of laterite soils passing sieve number 200 from the country of this research, Nigeria, gives a range of 10–60% [1,43–46], justifying why fewer research works are carried out using this soil. A current study by Osinubi et al. [47] examined the possible effects of calcite precipitate caused by Bacillus Coagulant on the geotechnical characteristics of lateritic soil for usage as road and embankment materials. They noticed that when the suspension density of *B. Coagulans* increased, so did their UCS values, signifying a counterclaim to MICP-improved soils with high fine particle percentages. Unfortunately, their recommendations were based on limited tests conducted (calcite content, unconfined compressive strength, and microanalysis test), which is insufficient to counter various researchers' claims on the difficulty of bacterial activity occurring in very fine soils. More experimental investigations are needed on the applicability of MICP to improve fine-grained soils. Laterites and the

lateritic soil used in Osinubi et al. [47] research are similar soils formed from the chemical weathering of crystalline rocks with a significant distinguishing difference in their molecular ratio [48].

The typical treatment practices used in biocementation may also be why there are fewer studies on applying MICP to enhance soil qualities for road construction. In the bid to achieve homogeneity in MICP-improved soil samples, various treatment approaches practised by researchers include treatment by immersion [49–53], surface percolation method [49,54,55], and pressure injection method [56,57]. These methods may not apply to road construction, as most of the time, soil improvements are performed at the soil's optimum moisture content (i.e., controlled moisture content) to achieve the best results. The premixing method [58], which has not been used frequently in previous MICP research projects, is the most appropriate for improving soil for road construction. Thus, using the premixing technique to mimic the methods used in road construction, this study carried out a systematic experimental study to investigate the use of the bacterium *Bacillus Anthracis* for laterite subgrade soil improvement for road construction.

## Materials and methods

### Materials

**Laterite soil.**  The natural ecosystem of three laterite soils was altered to promote the development of microorganisms, i.e., biostimulation [25,59]. The sites from which the soils were collected were at Canaan City in Canaan Land, Ota, Ogun State, Nigeria (Long. 6° 39'27.76"N, Lat. 3° 9'22.58" E), a borrow pit at Iju Ogun State, Nigeria (Long. 6°40'52.19"N, Lat 3°09'10.3"E), and a borrow pit at Atan, Ota, Ogun State, Nigeria (Long. 6°40'23.56"N, Lat 3°07'02.37"E).

**Microorganism.**  Three urease-positive bacteria were successfully cultivated from the soil sample taken from Canaan City in Canaan Land, Ota, Ogun State; however, no urease-positive bacteria were detected in the other two soil samples. The biochemical method was used to culture the following microorganisms from the soil sample: *Bacillus Anthracis* (ATCC 14578), *Klebsiella* (ATCC 10031), and *Bacillus Lentus* (ATCC strain 10840).

**Cementation reagent.**  The cementation reagent was prepared using urea ($CH_4N_2O$), Calcium chloride ($CaCl_2$), sodium bicarbonate ($Na_2CO_3$), and Nutritional broth sourced from a chemical market in Lagos, Nigeria.

**Bacteria suspension.**  Three (3) grams of nutritional broth and twenty (20) grams of urea were added to one litre of distilled water to create the bacterium suspension used to inoculate the bacteria. The MacFarland scale [60] was used to modify the turbidity of the bacterial suspensions.

### Methods

**Experimental procedure.**  Fig 1 shows a schematic step-by-step process employed in this research. Before the study was carried out, the protocols used in this research underwent executive review and were adjudged to have qualified for a waiver by the full committee of the Covenant Health Research Ethics Committee (CHREC). Consent and verbal approval were also obtained from the Department of Civil Engineering, College of Engineering, Covenant University's leadership on the quasi-field experiment carried out within the department.

**Isolation of the bacterium species.**  The bacteria were isolated at the microbiology laboratory at Covenant University, Ota, Ogun State. One hundred millilitres of distilled water was placed inside a sterile bottle. One gram of the soil sample was added to the distilled water, the container was closed, and the mixture was shaken vigorously. Using a pipette, 1 ml of solution was added to a Petri plate. Fifteen millilitres of nutrient agar was added to the plate, covered with a lid, and gently agitated to ensure the agar covered the whole surface. The plates were incubated for a minimum of 24 hours and a maximum of five days after being flipped over. The plates were removed from the incubator, and the number of bacterial colonies between 20 and 300 was determined.

**Culture medium and growth conditions.**  This study followed the methods published by Stocks-Fischer et al. [61] regarding growth conditions and bacterial culture medium.

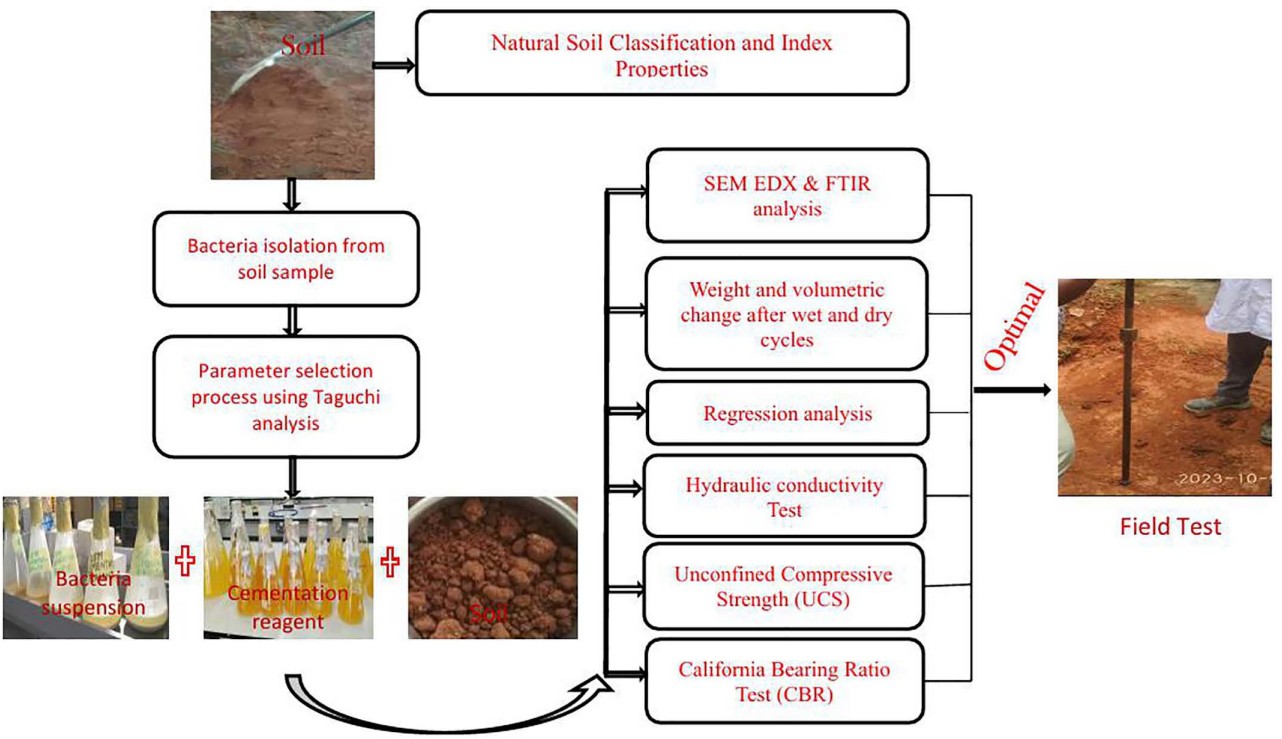

**Fig 1. Schematic step-by-step process employed in this research work.**

**Parameter selection process.** Graham [62] stated that the characteristics of soil that should be measured for a road construction project should be in one of four categories: soil classifications, strength identification, stiffness or compressibility identification, and hydraulic conductivity. Using the Taguchi design of experiments, a design of experiment, TDoE, was created to determine the best parameters data points to employ to examine how biocementation would change the characteristics of laterite soil based on these four criteria. The parameters considered for this study, at their various ranges based on past research works, are:

a) eleven (11) MacFarland scales of bacteria concentration [60],

b) bacteria: *Bacillus anthracis* (ATCC 14578), *Bacillus Lentus* (ATCC strain 10840), *Klebsiella* (ATCC 10031),

c) molar concentration of cementation reagent; 0.25, 0.5, 0.75, 1, and 1.25 molar [18,63,64],

d) ratio of bacteria to cementation reagent; 1:4, 2:3, 1:1, 3:2, and 4:1,

e) time; 0, 4, 8, 12, and 16 hours before mixing [52,65–69],

f) temperature: 15, 20, 25, 30, and 35 °C [70].

The rapid titration method was used to run a calcite content test following the TDoE protocols. Results obtained were analysed using Taguchi analysis in the Minitab statistical tool (2021 version). Optimal points obtained from the analysis were used to run further tests.

**Natural soil classification and index properties.** Table 1 shows tests carried out on the natural soil to obtain its index properties. The results from these tests were used to characterise the soil based on the Unified Soil Classification System (USCS) and the American Association of State Highway and Transportation Officials, AASHTO [71].

**Table 1. Test carried out on natural soil.**

| TEST | CODE/PROCEDURES [72] |
|------|----------------------|
| Natural Moisture Content | BS 1377 part 2. |
| Particle Size Distribution | BS 1377 part 2. |
| Specific Gravity | BS 1377 test 6B |
| Atterberg Limits | BS 1377 test 1(A) |
| Compaction Characteristics | BS 1377 part 2 and BS 1924 |

**Tests carried out on natural soil and *Bacillus Anthracis* improved soil.** The following tests were carried out on the *Bacillus Anthracis* improved soil:

**California bearing ratio (CBR).** The test protocol described in ASTM D1883-16 [73] was used for this test. The California bearing ratio test, which gauges the soil's resistance to a standard plunger under controlled density and moisture conditions at a 1.25 mm/min pace, was applied to the untreated and *Bacillus Anthracis*-treated soil. A plunger was seated into the untreated and bacteria-treated soil with 50 N of force. The plunger kept going through the soil at a constant rate of 1.25 mm/min while the forces were recorded at 0.25 mm intervals. The entire penetration was limited to 7.5 mm. The CBR at 2.5 mm and 5 mm penetration was calculated using Equations 1 and 2, and the highest value was chosen as the CBR.

$$CBR @2.5 = \frac{load\ reading\ (kg)}{1370\ kg} \times 100$$

(1)

$$CBR @5.0 = \frac{load\ reading\ (kg)}{2055\ kg} \times 100$$

(2)

**Unconfined compressive strength (UCS).** The unconfined compressive strength (UCS) test was carried out per the test protocols described in ASTM D2166 [74] to ascertain the load per unit area at which the naturally occurring and bacterially treated soil falls in compression. The natural soil and *Bacillus Anthracis* soil specimen were positioned between the end plates of a load frame. A steel ball was put on the bearing plate. The steel ball and a proving ring were placed on the same line by adjusting the specimen's centre line. A dial gauge was fixed to measure the specimen's vertical compression. The load 'frame's gear position was changed to provide the appropriate vertical displacement. After applying the load, readings from the compression and proving dials were noted at 5 mm intervals. Loading was maintained until the failure was finished. The unconfined compressive strength of natural and *Bacillus Anthracis* improved soil was calculated using Equation 3.

$$q_u = \frac{p}{A}$$

(3)

Where

p = axial load when the soil fails,

A = corrected area = $\frac{A_o}{1-\varepsilon}$,

$A_o$ = the initial area of the soil specimen,

ε = axial strain = change in length/original length.

**Coefficient of permeability (k).** The test protocol described in BS 1377 [72] was used to compute the permeability coefficient. The falling head permeability test was used in this investigation. The permeability coefficient (k) was calculated using Equation 4.

$$k = \frac{2.303aL}{At} \log \frac{H_1}{H_2}$$

(4)

Where:

k = coefficient of permeability (m/s),

a = cross-sectional area of the standing pipe or tube ($m^2$),

A = diameter of mould used (m), L = length of specimen (m),

T = time seconds(s),

$H_1$ and $H_2$ = heights of standpipe initial and final water levels (m).

**Regression analysis.** To determine the relationship between the three categories of properties of the *Bacillus Anthracis* treated soil, according to Graham [62], a regression analysis was carried out using Microsoft Excel (2016). Since the ultimate requirement for any constructed road is strength, the CBR soil property was the dependent variable, while permeability and UCS were the independent variables. Bacteria-specific linear equations were generated to predict the CBR properties. Regression statistic tables were generated from the regression analysis to obtain multiple R, R-square, and adjusted R-square values. Analysis of variance, ANOVA, and tables were also generated to get the degrees of freedom and significance of F values. The coefficient of multiple linear regression was generated to obtain standard error and P values.

**Volumetric shrinkage and loss of weight.** The test protocol described in ASTM D559/D559M-15 [75] was used for this test. A volumetric shrinkage test was performed to understand the shrinkage behaviour of the compacted natural soil and the *Bacillus anthracis-treated* soil. Cylindrical specimens that were compacted and extruded were employed in this test. Fig 2 shows samples used for the volumetric shrinkage and weight loss test.

The cylindrical extrusions underwent 12 wetting and oven-drying cycles at a consistent temperature of 70 degrees Celsius. Every 48 hours, the diameter and height of each specimen were measured using an electronic digital Vernier Caliper, which was accurate to within 0.01 mm. The volume was calculated. The weight of each sample was obtained with a weighing balance. The volumetric shrinkage strain was computed using Equation 5 based on the average heights and diameters.

$$VSS = \frac{(V_o - V_f)}{V_o} \times 100$$

(5)

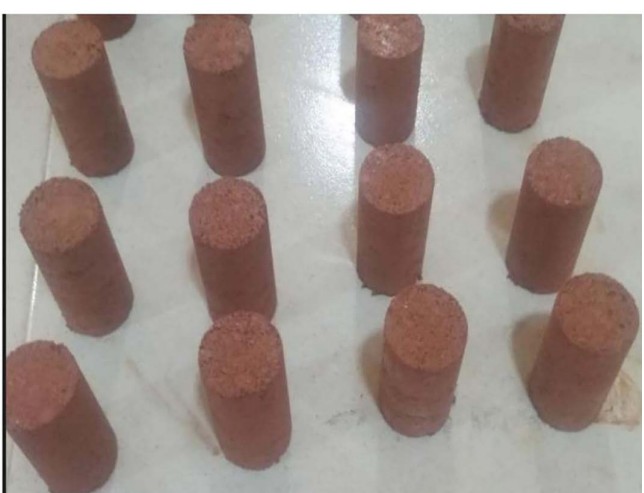

**Fig 2. *Bacillus Anthracis* treated soil samples used for the volumetric shrinkage and weight loss test.**

Where:

VSS = Volumetric shrinkage strain.

$V_o$ = Original volume of specimen.

$V_f$ = Final volume of specimen.

Equation 6 was used to calculate the percentage decrease in weight.

$$\% \ \textbf{\textit{descrease in weight}} = \frac{(\textbf{\textit{w}}_o - \textbf{\textit{w}}_f)}{\textbf{\textit{w}}_o} \times 100$$

(6)

Where:

$w_o$ = Original weight of the specimen.

$w_f$ = Final weight of a dry specimen.

**SEM and EDX analysis.** The soil morphology and elemental composition/chemical characterisation of the natural and *Bacillus Anthracis* treated soil was carried out using the scanning electron microscope. 2 mm of samples of the naturally occurring and *Bacillus Anthracis* treated soil were put on a double-stick tape and put on a sputtering machine for a minute and a half to give the sample a conductive quality. The samples were inserted into the scanning electron microscopy apparatus through a holder. Pictures were shot at a magnification of 2000x. Using a rotating knob on the SEM equipment, clear and appropriate pictures were taken. The samples were also subjected to energy dispersive X-ray (EDX) examination to determine the elemental distribution of the natural and *Bacillus anthracis*-treated soil.

**Fourier-transform Infrared Spectroscopy (FTIR).** FTIR spectra of samples were captured using Perkin-Elmer Spectrum RxI spectroscopy fitted with a Mullard I-alanine doped deuterated triglycine sulphate (DTGS) detector, covering a spectrum range of 4000–400 cm$^{-1}$.

**Quasi-field comparative test.** The field testing was conducted on a 1 × 1 m dug bed in a test bed. Two (2) sections made up the test bed. The optimal treated and natural soil were prepared at optimum moisture content (OMC), and each division received them. Fig 3a and 3b show the prepared test beds. In order to simulate a field subgrade thickness, Fig 3 required excavating and laying the 1 × 1 m structure at a depth of 300 mm. One division had the natural soil mixed at OMC, and the other the optimal bacterium-treated soil. Subsequently, a plate field compactor was employed to simulate the British standard heavy (BSH) compactive effort on both compartments. Care was made to ensure the treated soil portion had the same compaction runs as the natural soil section.

The in-situ CBR of the natural and optimally prepared soil was ascertained using the dynamic cone penetrometer (DCP). An in-situ density test using the core cutter method was also carried out on the two sections to check the field densities. Fig 4a illustrates the compaction procedure utilising the plate compactor, while Fig 4b–4d illustrates the DCP procedure to measure the soil's in-situ CBR.

The penetration per blow was calculated using Equation 7.

$$\textbf{penetration per blow} = \frac{\textbf{\textit{Penetration}} \ (\textbf{\textit{mm}})}{\textbf{\textit{total number of blows}}}$$

(7)

The penetration depth used was 300 mm, which is the depth of the test bed. The DCP hammer weighed eight (8) kilograms. A single common hammer factor was applied. The typical correlation between the CBR and DCP index was utilised to determine the CBR values of both the natural and treated soil.

## Results and discussion

### Index properties and natural soil classification

The index qualities of the natural laterite soil are listed in Table 2. The soil's colour was noted to be reddish-brown, which is indicative of laterite soils [76]. The American Association of State Highway Officials, AASHTO, [71] places the natural

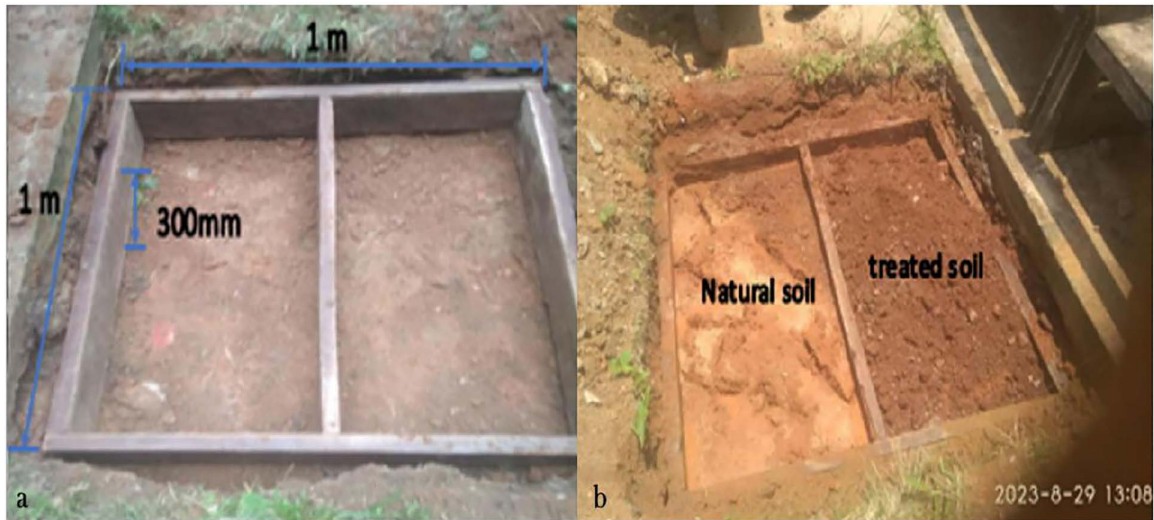

**Fig 3. Preparation of test bed and laying of compacted soil.**

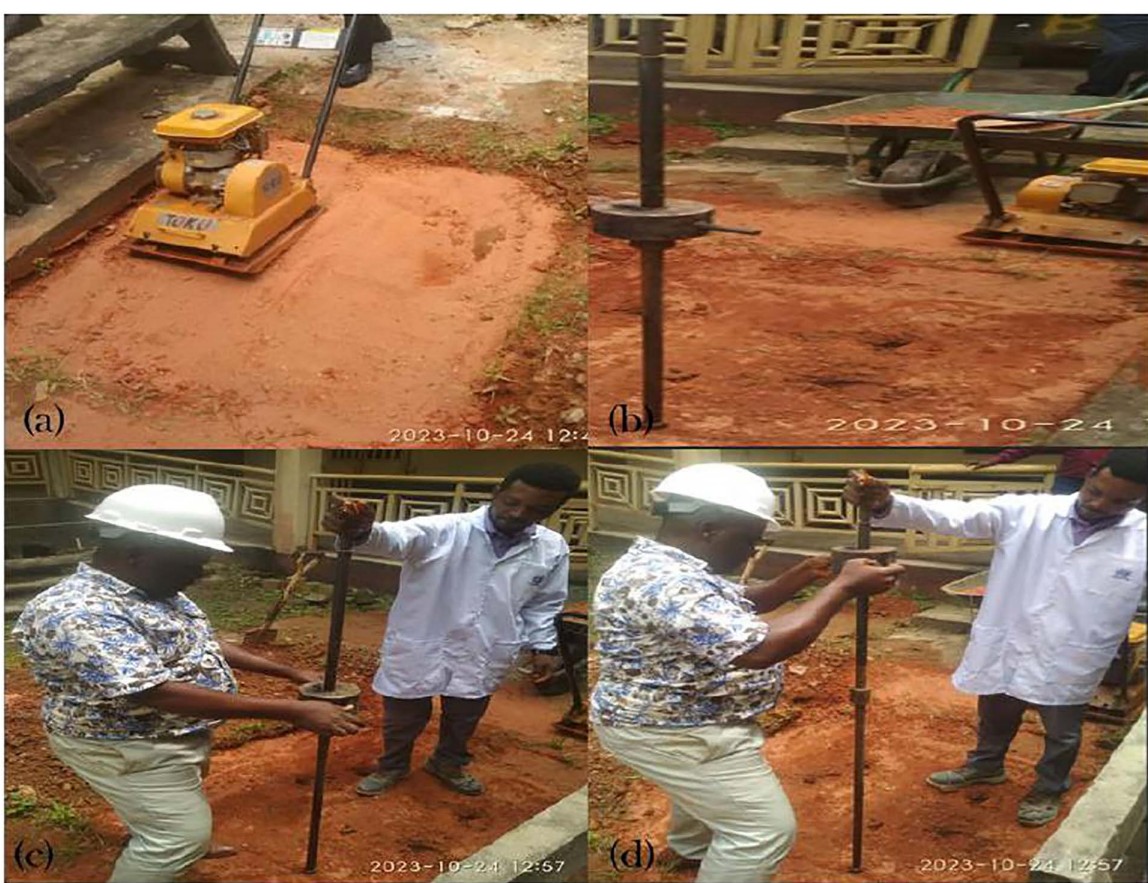

**Fig 4. a) Compaction of test bed, b – d, soil testing using the Dutch Cone penetrometer.**

**Table 2. Index properties of the natural soil.**

| Property | Quantity |
|---|---|
| Quantity passing No. 200 Sieve (%) | 52.54 |
| Natural Moisture Composition, (%) | 16.6 |
| Plastic Limit, (PL) % | 32.5 |
| Liquid Limit (LL) % | 17.3 |
| Plasticity Index (PI) % | 15.2 |
| Specific Gravity | 2.54 |
| AASHTO Classification | A – 6 (5) |
| USCS | CL |
| % gravel | 0.68 |
| % sand | 46.78 |
| % silt | 44.54 |
| % clay | 8% |
| Unconfined compressive strength (UCS) (kN m$^{-2}$) | 145 |
| CBR (unsoaked) (%) | 4 |
| CBR (soaked) (%) | 3.33 |
| Maximum Dry Density (MDD), Mg m$^{-3}$ | |
| British Standard Light (BSL) (Mg m$^{-3}$) | 1.56 |
| British Standard Heavy (BSH) (Mg m$^{-3}$) | 1.76 |
| Optimum Moisture Content (OMC) | |
| British Standard Light (BSL) | 13.2% |
| British Standard Heavy (BSH) | 12.7% |
| Physical appearance | Reddish Brown |
| Presiding Clay Mineral | Kaolinite |

soil in category A–6 (5). AASTHO describes soils in this category as fair to poor soils for use as subgrade materials, justifying the need for improvement. In addition, the soil is classed as a low plasticity clay (CL) by the Unified Soil Classification System (USCS).

The natural soil was further characterised using a wet sieve analysis, as shown by the particle size distribution curve in Fig 5.

The percentages of gravel, sand, silt, and clay in the natural laterite soil calculated from Fig 5 are 0.68, 46.78, 44.54, and 8%, respectively.

Fig 6 shows the X-ray diffraction analysis (XRD) used to determine the dominant clay mineral present in the soil. Quartz is the most common mineral with the highest peak and the most occurrences at various 2theta angles, consistent with research findings on the most common clay mineral in laterite [77].

Kaolinite is the primary type of clay mineral identified by XRD in the soil. The presence of kaolin indicates that the soil is less likely to undergo excessive swelling and shrinking when in contact with water [78].

Table 3 presents the oxide composition of the natural soil obtained from the X-ray fluorescence (XRF) analysis. This test served as a validation test for the soil type employed in the study. In geotechnical engineering, soils referred to as laterites have a molecular ratio of silica oxide ($SiO_2$) to sesquioxide (iron oxide, $Fe_2O_3$, and aluminium oxide, $Al_2O_3$) below 1.33, as expressed in equation 8.

$$\frac{SiO_2}{R_2O_3} = \frac{SiO_2}{Fe_2O_3 + Al_2O_3} < 1.33$$

(8)

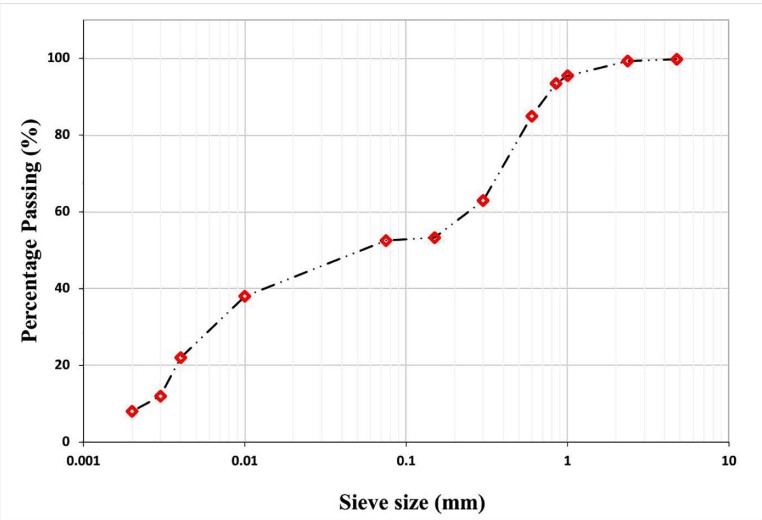

**Fig 5. Distribution curve for particles of the natural soil.**

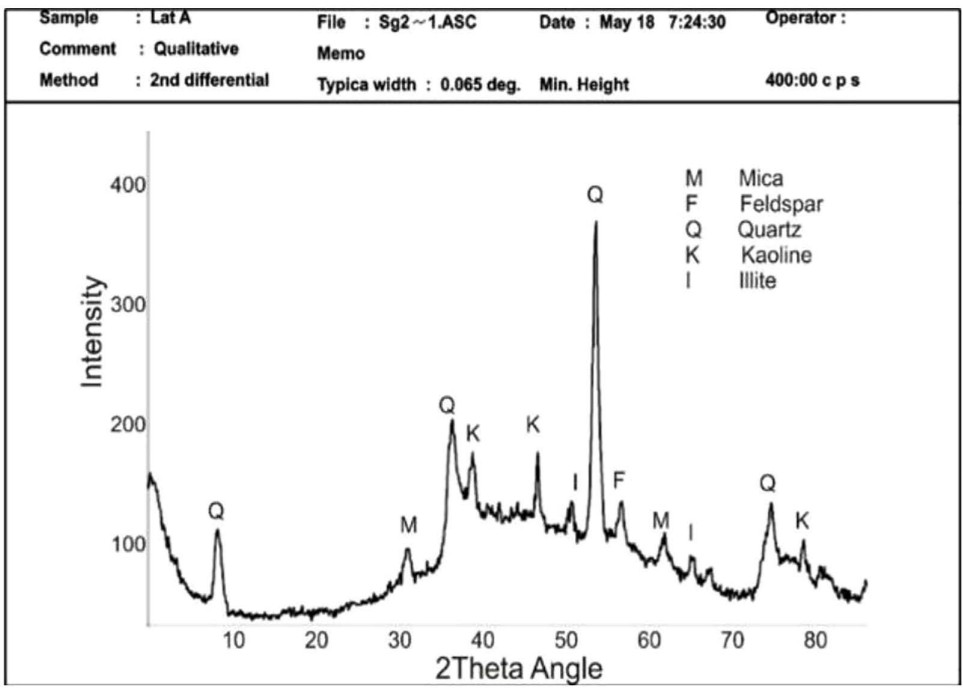

**Fig 6. Laterite soil XRD findings.**

The results showed that the content of iron oxide ($Fe_2O_3$) was 4.72%, silicon oxide ($SiO_2$) was 51.04%, and aluminium oxide ($Al_2O_3$) was 51.04%. Substituting these values into equation 8;

$$\frac{SiO_2}{Fe_2O_3 + Al_2O_3} = \frac{51.04}{4.72 + 34.44} = \frac{51.04}{39.16} = 1.3$$

**Table 3. Laterite soil oxide composition.**

| Oxide (%) | Composition |
| --- | --- |
| $Al_2O_3$ | 34.44 |
| $SiO_2$ | 51.04 |
| $fe_2O_3$ | 4.72 |
| $CaO$ | 1.75 |
| $MnO$ | 0.08 |
| $K_2O$ | 0.98 |
| $P_2O_5$ | 0.04 |
| $TiO_2$ | 1.25 |
| $MgO$ | 4.50 |
| $Na_2O$ | 0.90 |
| $Cr_2O_3$ | 0.22 |
| $SO_3$ | 0.08 |
| LOI | 0.02 |

The ratio being smaller than 1.33 suggests that the soil sample used in this study is a true laterite.

## Results of data points for selected parameters used in this study

**Calcite content results.** Several investigations have revealed that the concentration of bacteria used is a crucial component of the MICP process [79–81]. Higher bacterial concentrations have been reported to enhance calcite precipitation; however, Table 4 results show that the highest percentage of calcite was obtained at a low concentration of 3.0 x $10^8$ cells ml$^{-1}$ (experiment No. 2), which is in line with the conflicting findings of Shahrokhi-Shahraki [82] and Cheng [83]. The results of Soon et al. [24], who saw an increase in their soil's unconfined compressive strength, UCS, when they increased their molar concentration from 0.25 to 0.5 M, are in line with the cementation reagent's molar concentration (0.5 M) obtained in experiment 2.

**Taguchi analysis results.** The calcite content results were further analysed using the Taguchi analysis. The optimal parameter points from where the best soil improvement is expected according to the Taguchi analysis were obtained and presented in Fig 7. The plot of the mean and the mean of the signal-to-noise ratio of the optimal levels for each parameter is presented in Fig 7. These plots informed the use of the bacterium *Bacillus Anthracis* and all data points for each parameter used for this research work. The Taguchi parameter procedure also helped to minimise the number of experiments required if all data points for all parameters were considered.

The optimal parameter data points were obtained as peak points of graphs of each parameter in Fig 7 and used as protocols for subsequent tests in this research work according to the Taguchi analysis. The bacteria type Bacillus Anthracis, the bacteria concentrations: 3.0 x $10^8$, 6.0 x $10^8$, 15 x $10^8$, and 24 x $10^8$ cells ml$^{-1}$, the molar concentration of cementation reagent: 0.5 M, the mix ratio of bacteria with cementation reagent: 2:3, and the time before mixing: 4 hours are peak data points obtained for each parameter.

## Test carried out on *Bacillus Anthracis* improved laterite soil

After achieving the first prerequisite established by Graham [62] for soil to be used for road construction projects, namely soil classification, the California bearing ratio (CBR) test was carried out to determine the strength of the natural and modified soil.

**California bearing ratio (CBR) results.** Subgrade soils are categorized by the Overseas Road Note 31, TRRL [84], and the Nigerian Federal Ministry of Works [85] as S1, S2, S3, S4, S5, and S6 with CBR values of 2, 3–4, 5–7, 8–14,

**Table 4. Results of calcite contents after titration.**

| Experiment number | Bac-teria | Bacteria Conc. (x $10^8$ cells ml$^{-1}$) | Molar Conc. (g/l) | Bacteria +reagent | Time (h) | Sample reading Trial 1 | Sample reading Trial 2 | Average Sample reading | % CaCO$_3$ |
|---|---|---|---|---|---|---|---|---|---|
| 1 | A | 1.5 | 0.25 | 20/80 | 0 | 34.9 | 38.4 | 36.65 | 4.75 |
| 2 | A | 3.0 | 0.50 | 40/60 | 4 | 41.3 | 43.0 | 42.15 | 11.63 |
| 3 | A | 6.0 | 0.75 | 50/50 | 8 | 39.5 | 40.3 | 39.90 | 8.81 |
| 4 | A | 9.0 | 1.00 | 60/40 | 12 | 38.0 | 38.8 | 38.40 | 6.94 |
| 5 | A | 12 | 1.25 | 80/20 | 16 | 38.8 | 39.5 | 39.15 | 7.55 |
| 6 | B | 1.5 | 0.50 | 50/50 | 12 | 39.2 | 39.1 | 39.15 | 7.55 |
| 7 | B | 3.0 | 0.75 | 60/40 | 16 | 36.9 | 38.1 | 37.50 | 5.81 |
| 8 | B | 6.0 | 1.00 | 80/20 | 0 | 38.0 | 37.2 | 37.60 | 5.94 |
| 9 | B | 9.0 | 1.25 | 20/80 | 4 | 37.5 | 37.5 | 37.50 | 5.81 |
| 10 | B | 12 | 0.25 | 40/60 | 8 | 38.2 | 38.5 | 38.35 | 6.88 |
| 11 | C | 1.5 | 0.75 | 80/20 | 4 | 36.6 | 37.2 | 36.90 | 5.06 |
| 12 | C | 3.0 | 1.00 | 20/80 | 8 | 39.5 | 36.0 | 37.75 | 6.13 |
| 13 | C | 6.0 | 1.25 | 40/60 | 12 | 37.3 | 35.6 | 36.45 | 4.50 |
| 14 | C | 9.0 | 0.25 | 50/50 | 16 | 39.4 | 36.1 | 37.75 | 6.13 |
| 15 | C | 12 | 0.50 | 60/40 | 0 | 36.9 | 36.1 | 36.50 | 4.56 |
| 16 | AB | 1.5 | 1.00 | 40/60 | 16 | 36.0 | 35.6 | 35.8 | 3.69 |
| 17 | AB | 3.0 | 1.25 | 50/50 | 0 | 35.7 | 35.0 | 35.35 | 3.13 |
| 18 | AB | 6.0 | 0.25 | 60/40 | 4 | 34.4 | 36.0 | 35.20 | 2.94 |
| 19 | AB | 9.0 | 0.50 | 80/20 | 8 | 34.3 | 35.2 | 34.75 | 2.38 |
| 20 | AB | 12 | 0.75 | 20/80 | 12 | 33.9 | 34.0 | 33.95 | 1.38 |
| 21 | AC | 1.5 | 1.25 | 60/40 | 8 | 32.0 | 34.5 | 33.25 | 0.50 |
| 22 | AC | 3.0 | 0.25 | 80/20 | 12 | 34.5 | 33.1 | 33.80 | 1.19 |
| 23 | AC | 6.0 | 0.50 | 20/80 | 16 | 35.3 | 35.2 | 35.25 | 3.00 |
| 24 | AC | 9.0 | 0.75 | 40/60 | 0 | 34.6 | 36.0 | 35.30 | 3.06 |
| 25 | AC | 12 | 1.00 | 50/50 | 4 | 34.7 | 32.8 | 33.75 | 1.13 |

Bacteria A = *Bacillus Anthracis*, B = *Klebsiella,* and C = *Bacillus Lentus*.

15–29, and >30%, respectively. The soaked CBR value of the laterite soil used in this research was 3.33%, similar to CBR values obtained by Etim et al. [44] and Sani, Etim & Joseph [86], who used laterite in their research work. Additionally, the Overseas Road Note 31, TRRL [84], and the Nigerian Federal Ministry of Works [85] said poor subgrade materials are soils with a CBR of less than 5%, suggesting that the natural soil employed is a poor subgrade material. Fig 8a demonstrates how the penetration vs. load reading for soil treated with different concentrations of *Bacillus Anthracis* varies. Fig 8b displays the calculated CBR values for *Bacillus Anthracis*-treated soil. The CBR was computed using Equations 1 and 2 following the acquisition of load data at 2.5 mm and 5 mm penetration.

Plotted curves in Fig 8a were found to be convexity upwards; therefore, no adjustments were required. Load reading of *Bacillus Anthracis* treated laterite soil at 2.5 mm for $3 \times 10^8$, $6 \times 10^8$, $9 \times 10^8$, $15 \times 10^8$, and $24 \times 10^8$ cells ml$^{-1}$ were obtained as 57.428, 36.332, 82.040, 29.300, and 58.600 kg, respectively. At 5 mm, load readings were obtained as 123.060, 55.084, 148.844, 55.084, and 87.900 kg, respectively. These values were inputted into Equations 9 and 10 to compute the CBR. The highest calculated CBR was adopted as the improved soil CBR and plotted in Fig 8b. Calculated CBR values for *Bacillus Anthracis* treated soil at $3 \times 10^8$, $6 \times 10^8$, and $9 \times 10^8$ cells ml$^{-1}$ gave better laterite soil strength improvements, with the best improvement obtained at $3 \times 10^8$ cells ml$^{-1}$ concentration. An 80% improvement in the CBR of the natural soil was observed at $3 \times 10^8$ cells ml$^{-1}$ treatment. Oyediran and Ayeni [87] gave reasons for increased CBR during MICP soil

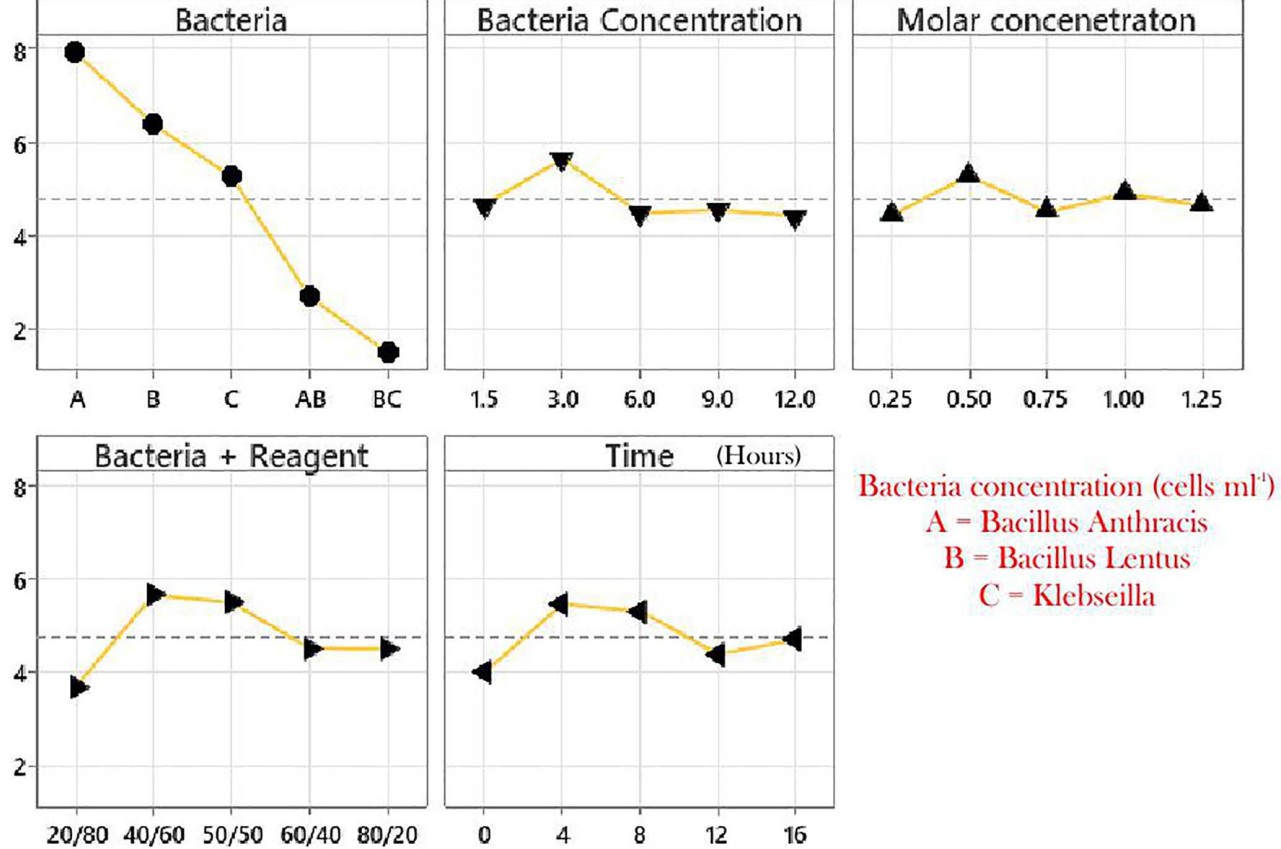

**Fig 7. Optimal points for each biocementation parameter.**

improvement due to a reaction that produced calcite, which bonds the soil particles, creating a denser soil matrix. Initially, the 3.33% CBR value obtained for the natural laterite soil was categorised in the S2 class by the Nigerian Federal Ministry of Works [85] and TRRL [84]. Improvements moved the soil from an S2 to an S3 class, indicating an enhanced subgrade strength.

**Unconfined compression strength results.** To assess the stiffness or compressibility properties of the Bacillus *Anthracis* treated soil [62], a careful search for global standards of the accepted UCS range for soils used as subgrade materials yielded the standard requirement of $800\,kN\,m^{-2}$ for Malaysia [88] and between $600–1,720\,kN\,m^{-2}$ for the Overseas Road Note 31, TRRL [84]. For the natural soil utilised in this study, the UCS value was $145\,kN\,m^{-2}$, which did not meet any of these standards, justifying the need for improvement. Fig 9a and 9b show the plot of the stress-strain relationship for the 7 and 28-day B – Anthracis modified soil.

The best strength values were obtained for the $3 \times 10^8$ cells $ml^{-1}$ concentration of B-Anthracis treated soil at 7 and 28 days. The UCS values obtained on these days were $379.11$ and $421.23\,kN\,m^{-2}$, respectively, corresponding to a 23.41% and 27.62% increase in the UCS value of the natural soil.

Table 5 shows equations of trendlines drawn along the linear regions used to obtain the stiffness in terms of Young's modulus of the modified soil. The stiffness of the soil was obtained as 3,786. Improvements in the stiffness property of the soil were obtained at all B – Anthracis concentration treatments. The best-improved soil with the highest stiffness was obtained at $3.0 \times 10^8$ cells $ml^{-1}$ *Bacillus Anthracis* treatment for both the 7 and 28-day UCS. The soil was improved from

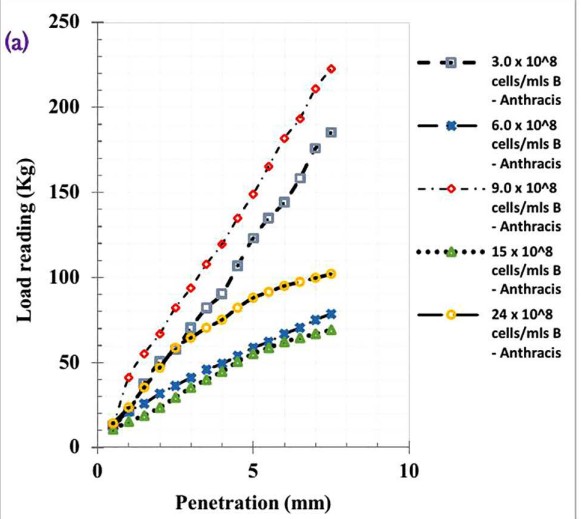
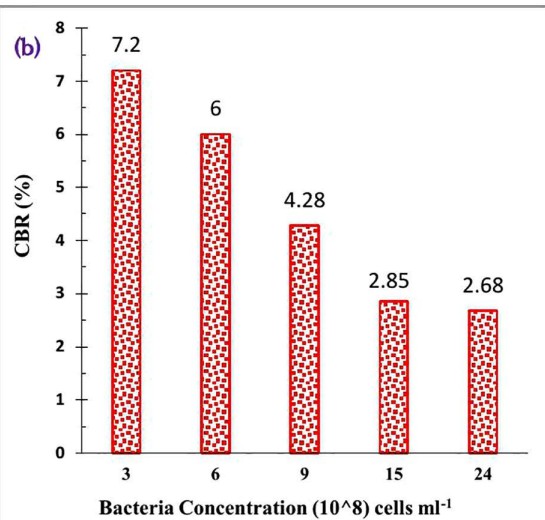

**Fig 8. a) Penetration vs. load reading for soil treated with different concentrations of _Bacillus Anthracis_ b) CBR values for _Bacillus Anthracis_ treated soil.**

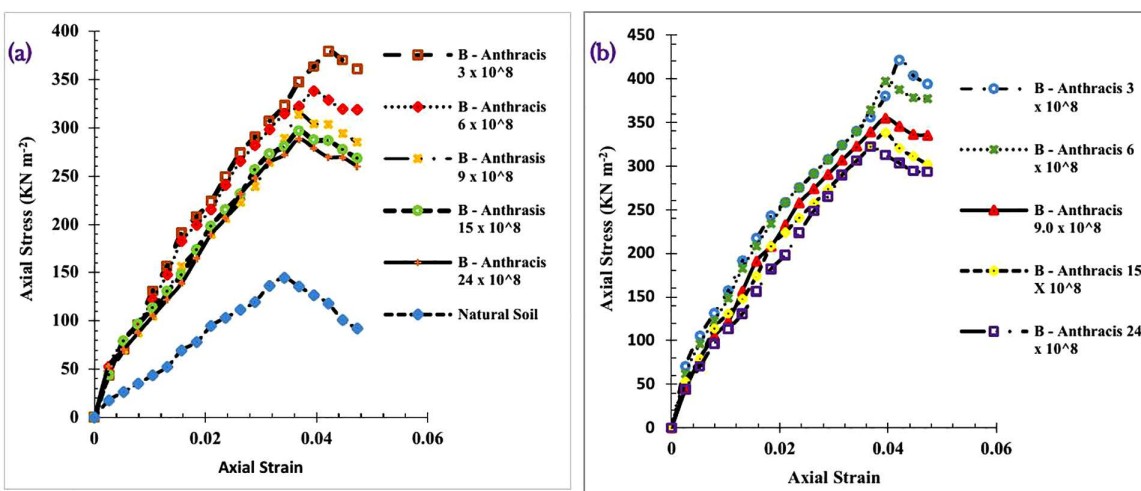

**Fig 9. a) Variation of stress-strain relationship for 7 days B – Anthracis modified laterite soil.** b) Variation of stress-strain relationship for the 28 days B – Anthracis modified laterite soil.

"very soft to soft" soil to "stiff to very stiff" soil [89] for all _B- Anthracis_ treatments. The $R^2$ goodness of fit values were used to confirm the validity and accountability of Young's modulus obtained. R-squared values greater than 90% were obtained for all concentrations.

Fig 10 is a multiple bar chart comparing the seven and twenty-eight-day UCS for _Bacillus Anthracis_ treatment at different concentrations. The seven-day UCS values were placed as the secondary axis in a clustered column, illustrating the variation in the results obtained after 7 and 28 days.

For all bacteria at varied concentrations, there was an overall significant improvement in strength or an increase in UCS values from 7 days to 28 days, an excellent soil quality required for road construction. The UCS value of the natural soil

**Table 5. Equations of trendlines drawn along the linear regions that were used to obtain the stiffness of the modified soil.**

| B – Anthracis Concentration (cells ml$^{-1}$) | Linear equation for seven (7) days (stiffness, kN m$^{-2}$) | Linear equation for twenty-eight (28) days (Stiffness, kN m$^{-2}$) |
|---|---|---|
| 3.0 x 10$^8$ | y = 11,709x + 6.3058, R² = 0.9956 (11, 709) | y = 14,261x + 17.898, R² = 0.9449 (14, 261) |
| 6.0 x 10$^8$ | y = 10,504x + 14.09, R² = 0.968 (10, 504) | y = 13,600x + 14.362, R² = 0.9598 (13, 600) |
| 9.0 x 10$^8$ | y = 10,954x + 7.2633, R² = 0.9571 (10, 954) | y = 11,615x + 9.0233, R² = 0.9711 (11, 615) |
| 15 x 10$^8$ | y = 10,618x + 10.763, R² = 0.9537 (10, 618) | y = 12,275x + 10.814, R² = 0.968 (12, 275) |
| 24 x 10$^8$ | y = 8,514.3x + 16.965, R² = 0.9353 (8,514) | y = 10,617x + 9.0163, R² = 0.972 (10, 617) |

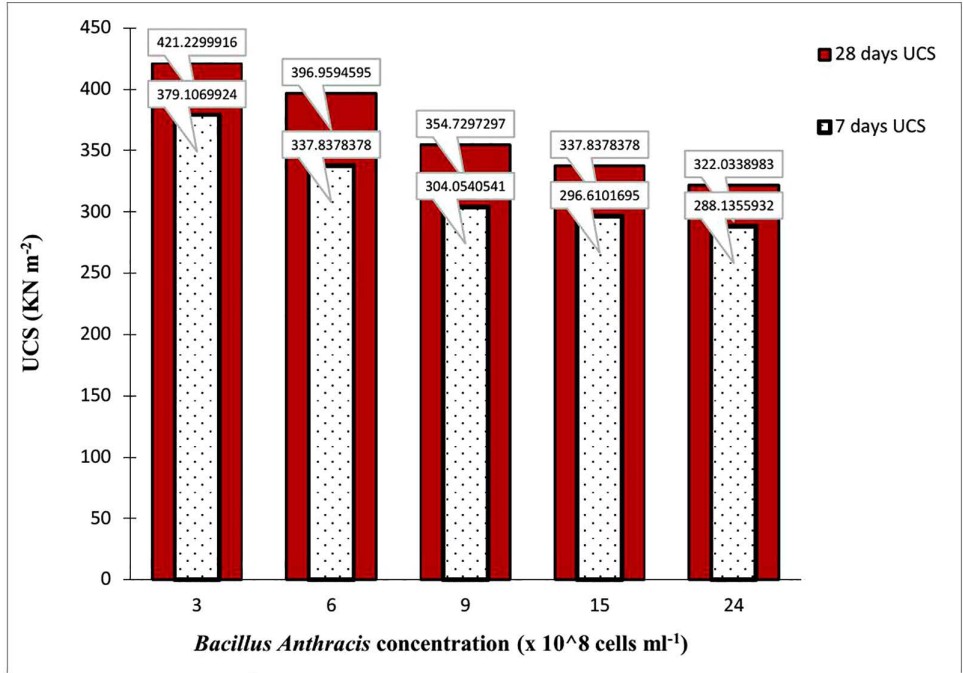

**Fig 10. Variation of the 7- and 28-day UCS values B- Anthracis modified soil.**

increased by 191% at 3.0 x 10$^8$ cells ml$^{-1}$ *Bacillus Anthracis* treatment, whereas increases of 174, 144, 133, and 122% were noted at 6.0, 9.0, 15, and 24 x 10$^8$ cells ml$^{-1}$, respectively. Results obtained for the USC values align with those obtained from the CBR test. The rise in UCS values is consistent with the results of Animesh and Ramakrishnan [90], who similarly found that applying MICP to their soil increased UCS values by 2.9 times. Similar patterns of rise in UCS were noted by Gowthaman et al. [91], Zhao et al. [92], and Choi et al. [93] in their respective research findings.

**Coefficient of permeability (k).** The permeability coefficient of the natural soil was obtained as 2.31368E-05 and 2.84389E-05 cm sec$^{-1}$ after compaction using British standard light (BSL) and British standard heavy (BSH) efforts, respectively. Several studies [94–99] have reported that the application of MICP causes the soil's coefficient of permeability to drop. The variance in *Bacillus Anthracis*-treated soils' coefficient of permeability when compacted with British standard light and British standard heavy is depicted in Fig 11. In agreement with research studies, there was an observed decrease in the coefficient of permeability of the natural soil when modified with the bacterium *Bacillus Anthracis* at all concentrations and using the British standard light (BSL) and the British standard heavy (BSH) compactive efforts. The best improvements were observed at the bacterium concentration of 3.0 x 10$^8$ cells ml$^{-1}$, which supported the results obtained from the CBR and UCS tests.

At 3 x 10$^8$ cells ml$^{-1}$ of *Bacillus Anthracis*-treated soil for BSL and BSH compaction, respectively, a 35% and 40% decrease in the coefficient of permeability of the natural soil was observed. For class S4 soils, the Nigerian Federal Ministry of Works and Housing [85] suggested a minimum subgrade thickness of 450 mm. If the soil utilized in this study had been utilized as a subgrade material for roads, compacted with light rollers, and kept wet all the time, it would have taken the following number of days for moisture to seep through the entire 450 mm thick layer of naturally compacted subgrade; $\frac{45}{0.0000284389} = 1,582,339.683 \ secs$ for moisture to seep through, i.e., 18.31 days for BSL compaction. For BSH compaction, it would take $\frac{45}{0.0000231368} = 1,944,953.494 \ secs$ or moisture to seep through, i.e., 22.51 days. Best improvements at 3 x 10$^8$ cells ml$^{-1}$ of *Bacillus Anthracis* -treated soil further increased the permeability/time taken for moisture to seep through the entire subgrade thickness to 28.54 and 37.39 days for BSL and BSH compactive effort, respectively. These improvements protect the subgrades from water infiltration.

**Regression analysis.** A regression analysis was carried out to establish a bacteria-specific relationship between the strength (CBR), compressibility (UCS), and hydraulic conductivity properties of the *Bacillus Anthracis*-treated soil for road construction. Equation 9 shows the relationship between the CBR, permeability using the BSH compactive effort, and seven-day UCS. Table 6 shows that the adjusted R-squared value from the regression statistics was 97.4%, indicating a good fit of the model.

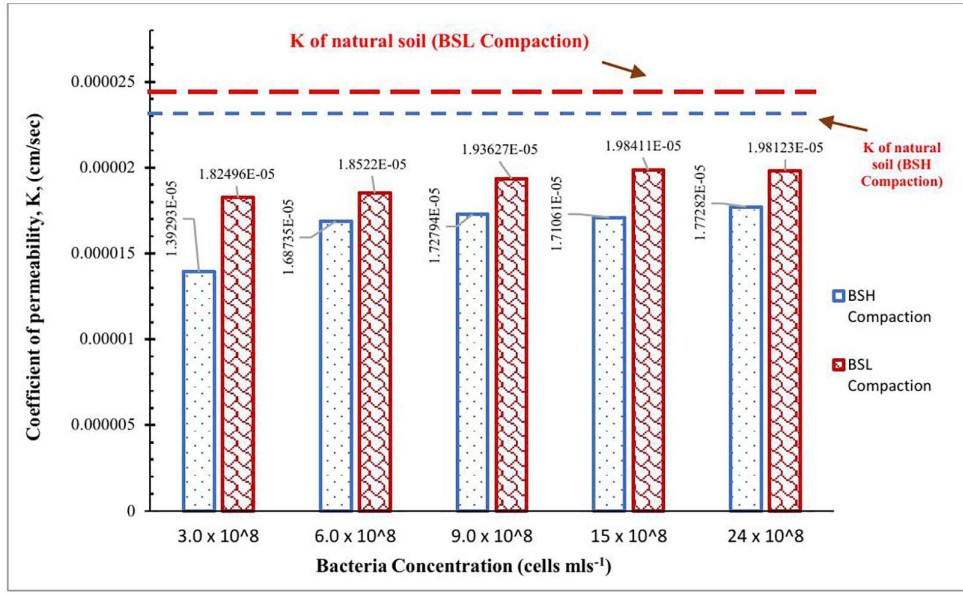

**Fig 11. Variation of the coefficient of permeability of B- anthracis modified soil.**

**Table 6. Regression statistics.**

| | |
|---|---|
| Multiple R | 0.993590525 |
| R Square | 0.987222131 |
| Adjusted R Square | 0.974444262 |
| Standard Error | 0.315410667 |
| Observations | 5 |

**Table 7. ANOVA.**

| | df | SS | MS | F | Significance F |
|---|---|---|---|---|---|
| Regression | 2 | 15.224 | 7.612 | 43.7886 | 0.022327129 |
| Residual | 2 | 0.348 | 0.173 | | |
| Total | 4 | 15.571 | | | |

**Table 8. ANOVA.**

| | Coefficients | Standard Error | t Stat | P-value | Lower 95% | Upper 95% | Lower 95.0% | Upper 95.0% |
|---|---|---|---|---|---|---|---|---|
| Intercept | −33.097 | 10.883 | −3.041 | 0.09 | −0.182 | 118.403 | −0.182 | 118.403 |
| 7 days UCS | 0.079 | 0.0151 | 5.209 | 0.035 | −0.064 | 0.057 | −0.064 | 0.057 |
| Permeability (BSL) | 745498.753 | 374463.059 | 1.991 | 0.184 | −5024272.809 | −553148.857 | −5024272.809 | −553148.857 |

$$\text{CBR} = 0.079\text{UCS (7 days)} + 745498.8k\text{ (BSH)} - 33.1 \qquad \text{Adj R}^2 = 97.8\% \tag{9}$$

The significance value, F, from the analysis of variance, ANOVA, table, Tables 7 and 8, is below 0.05, which signifies that Equation 9 is reliable. The predictive value, P, of permeability is above 0.05, indicating that the permeability of the *Bacillus Anthracis*-treated soil has little effect on strength. Hence, Equation 9 is now expressed as Equation 11.

$$\text{CBR} = 0.079\text{UCS (7 days)} - 33.1 \qquad \text{Adj R}^2 = 97.8\% \tag{10}$$

Equation 11 presents another relationship generated from the regression analysis with a significance value, F, and a predictive value, P, less than 0.05.

$$\text{CBR} = 59.11 - 2788710.83K\text{ (BSL)} \qquad \text{Adj R}^2 = 97.8\% \tag{11}$$

**Volumetric and weight change after wet and dry cycles.** The purpose of carrying out the wet and dry cycle test was to ascertain how much the treated compacted 'soils' pore pressure varied [100]. According to Ahenkorah et al. [99], this test is also a helpful indicator of how resistant the biotreated soil is to different environmental conditions. Fig 12a shows the volume change of the natural soil, while Fig 12b shows the volume change of *Bacillus Anthracis*-treated soil after 12 cycles. Fig 12c shows the weight change for the natural soil, while Fig 12d shows the weight change of *Bacillus Anthracis*-treated soil after 12 cycles.

Results obtained for the natural soil in Fig 12a showed a general decrease in volume after consequent wet and dry cycles. The volumetric shrinkage strain, VSS, calculated using Equation 5 for the natural soil, showed a 31.6% shrinkage in volume for the first cycle, 30.38, 17.3, 10 and 5.7% for the second, third and fourth cycles, respectively. Consequently, the volumetric shrinkage was constantly close to unity for the remaining cycles. A noticeable decrease in the VSS of the *Bacillus Anthracis* -treated soils was observed at all concentrations. The lowest VSS was at 3 x $10^8$ cells ml$^{-1}$ *Bacillus*

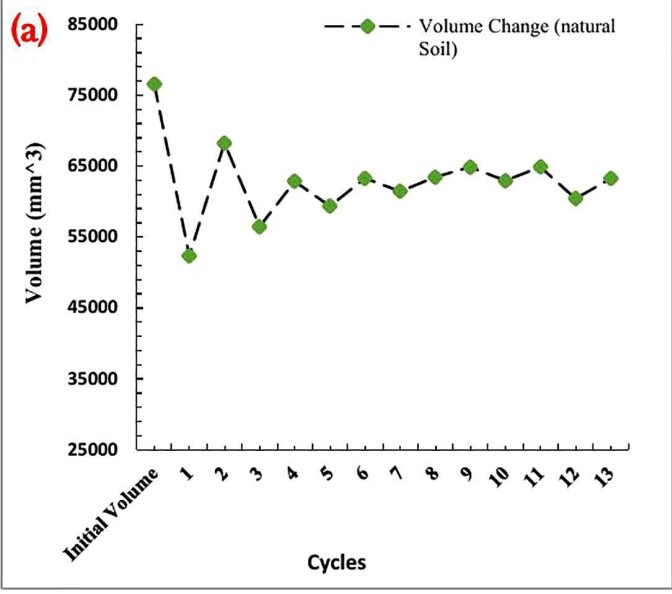

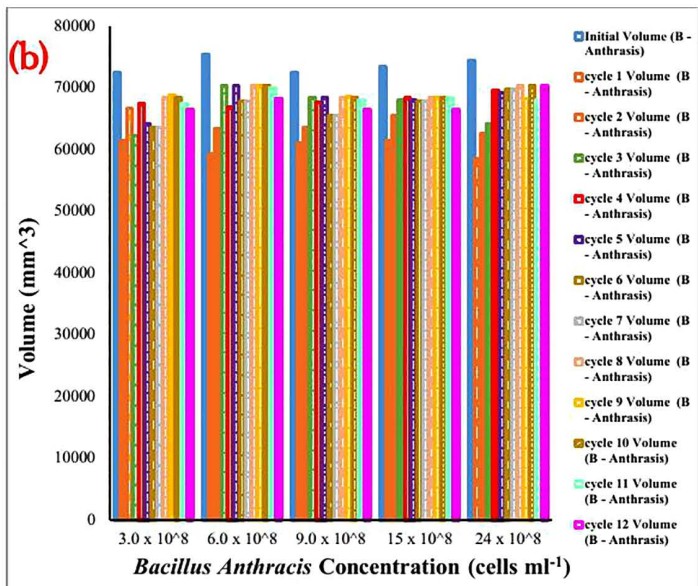

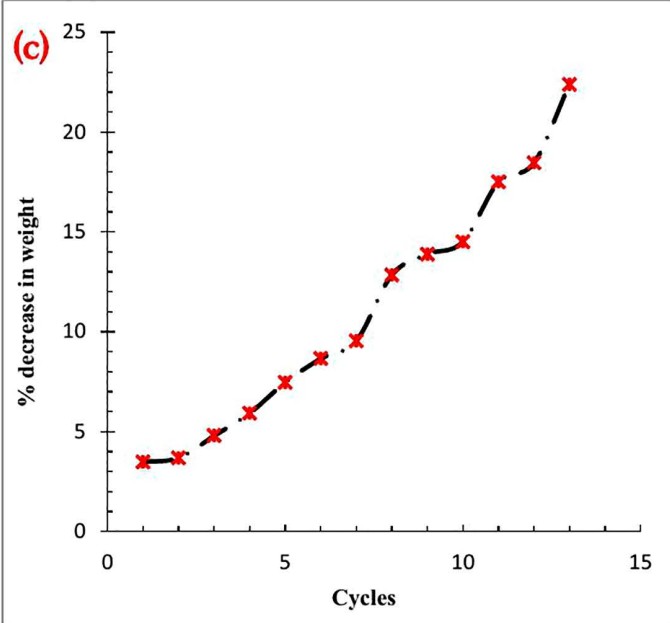

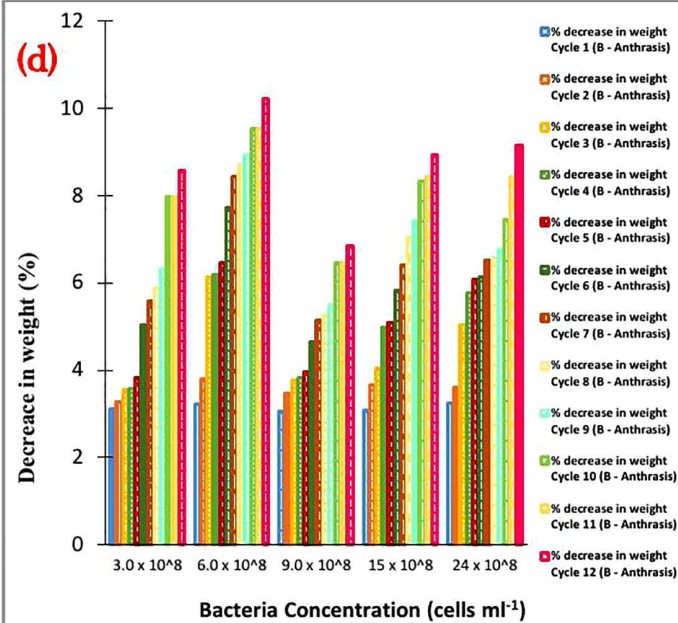

**Fig 12. A) Variation of volume change of the natural soil after 12 cycles, b) Variation of volume change of *Bacillus Anthracis* treated soil after 12 cycles, c) Variation of weight change for the natural soil after 12 cycles, d) variation of weight change of *Bacillus Anthracis* treated soil after 12 cycles.**

*Anthracis* treated soil, having VSS of 15, 8, 6, and 7% for the first, second, third, and fourth cycles, respectively, then a constant close to unity value for the remaining cycles. Research conducted by Camillis, Emidio, and Bezuijen [100] revealed a similar decline in the volume change of their natural and treated soil samples. Calcite formed a link between the soil particles and filled the pore spaces, producing a more durable subgrade material, which explains the commonly

seen decrease in volume change [101]. From as little as 3.7% after the first cycle to as much as 22% of its total weight after the twelfth wet and dry cycle, the natural soil experienced weight loss. Research by Ahenkorah et al. [99] and Gowthaman et al. [95] supports weight loss with increased wet and dry cycles. The percentage decrease in weight of the *Bacillus Anthracis* treated soil was less than that of the natural soil, which indicates the bonding of soil particles by the $CaCO_3$ crystals formed [102].

**Field testing.** For the field study, $3 \times 10^8$ cells ml$^{-1}$ of *Bacillus Anthracis* was the optimum bacterial concentration. The number of blows recorded for the natural soil at the first trial was five, which drove the DCP rod in by 300 mm. The penetration per blow was obtained as follows;

$$\frac{Penetration\ (mm)}{total\ number\ of\ blows} = \frac{300}{5} = 60\ mm/blows$$

The ' 'DCP's hammer weighed 8 kg, so the standard hammer factor of one was used.

$$DCP\ index\ in\ mm/blow\ =\ 60\ x\ 1\ =\ 60\ mm/blow$$

Table 9 shows two trials of in-situ density and CBR results. The treated soil performed better when the laboratory strength results were compared to the field strength. A significant distinction was that, in contrast to the S3 class obtained at the laboratory scale, the field treated soil was categorised as S4 (CBR 8%) by the Nigerian Federal Ministry of Works [85] and the Overseas Road Note 31, TRRL [84], indicating better field performance.

**Microstructural results.** Fig 13(a) shows the natural ' 'soil's SEM image and EDX results. Fig 13(b) shows the SEM image and EDX results for *Bacillus Anthracis* improvement at $3 \times 10^8$ cells ml$^{-1}$.

Numerous micro pores and fissures in the natural soil's SEM images were observed at a 2000x magnification, as shown in Fig 13a. Similar observations of micropores and cracks were made in a research work by Awoyera et al. [103]. The EDX image of the natural soil in Fig 13a showed a high percentage of oxygen. Additionally, sodium, nitrogen, silicon, aluminium, and chloride were observed. The SEM images for the $3.0 \times 10^8$ cells ml$^{-1}$ *Bacillus Anthracis* treated soil in Fig 13b revealed some pore gaps even following improvement, which could be attributed to the aggregation of soil particles due to the MICP modification process. In line with the results obtained by Osinubi et al. [20] and Gowthaman et al. [104], calcite crystals were also detected. A significant proportion of oxygen was seen in the $3.0 \times 10^8$ cells ml$^{-1}$ *Bacillus Anthracis* treated soil. Carbon was also seen, suggesting the presence of carbonate ($CO_3^{2-}$) ions in calcite, corroborating the findings from the SEM image about the presence of calcite.

**Fourier-transform Infrared Spectroscopy (FTIR).** Fig 14 shows the Fourier Transformation Infrared spectroscopy (FTIR) results used to characterize the mineralogical and geochemical properties of natural soil, while Fig 15 shows the Fourier Transformation Infrared spectroscopy (FTIR) results for *Bacillus Anthracis* treated soil.

From the infrared spectroscopy correlation tables, a slight peak of C=O bonds and O-H (acids) bonds was observed in the functional group regions, while in the fingerprint region, possible peaks of =C-H & =CH2 and O-H bending were seen. Etim et al. [44] and Chipera and Bish [105] reported that peaks at a wavelength of 3652.8 cm$^{-1}$ might depict traces of clay minerals present in the natural soil sample. Characteristic peaks of Al-OH bending and SiO stretching vibrations of

**Table 9. Test results of in-situ density and CBR for the natural soil and optimally treated soil.**

| Natural Soil | | | | | Treated Soil | | | |
|---|---|---|---|---|---|---|---|---|
| Moisture content (%) | | In-situ dry density (Mg m$^{-1}$) | Number DCP blows | In-situ CBR (%) | Moisture content | In-situ dry density (Mg m$^{-1}$) | Number DCP blows | In-situ CBR (%) |
| BSL | 17.3 | 1.61 | 5 | 3 | 16.9% | 1.68 | 9 | 6 |
| BSH | 15.5 | 1.74 | 8 | 5 | 15% | 1.86 | 12 | 8 |

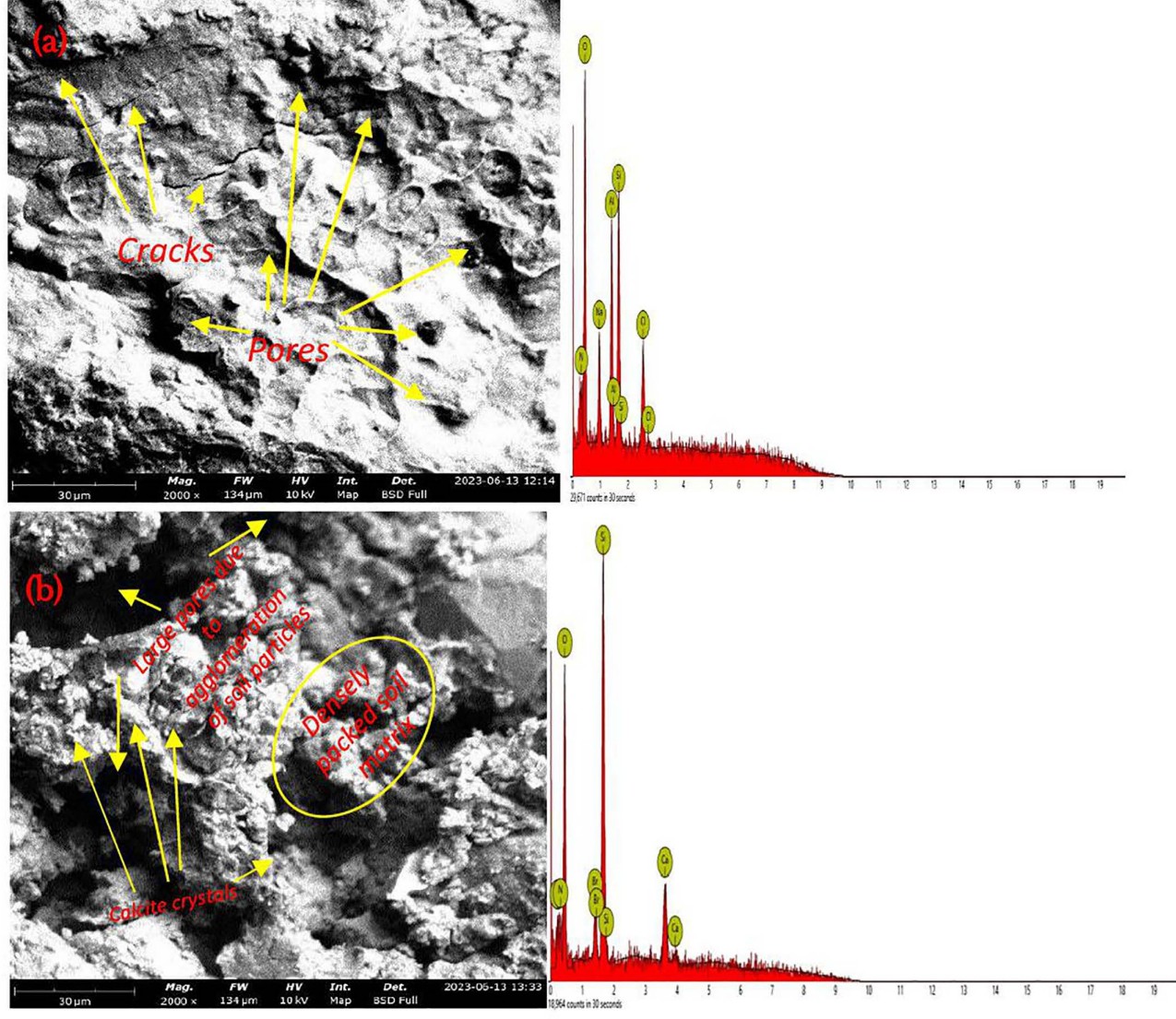

**Fig 13. A) SEM image and EDX results for the natural soil; b) SEM image and EDX results for *Bacillus Anthracis* soil improvement at 3 x 10^8 cells ml^-1.**

weathered sheet silicates, principally illite and kaolinite, were also observed in the FTIR results of the natural soil, confirming the presence of Kaolinite minerals as observed from the XRD results [103–107]. Characteristic quartz double peak 778−797 cm^-1 and perpendicular vibrations of silicate minerals 693 cm^-1 Si-O were also observed [103–107]. Veerasingam and Venkatachalapathy [107] reported three distinct IR absorption peaks for calcium carbonate between 400 and 4000: 1460 cm^-1, 880 cm^-1, and 712 cm^-1. The FTIR results for the *Bacillus Anthracis*-treated soil showed a peak of 1457.4 cm-1, confirming the presence of calcite and the MICP reaction.

## Conclusion

This study investigated the effects of the bacterium *Bacillus Anthracis* on laterite subgrade soil improvement for road construction. Based on the experimental findings, the following conclusions were drawn;

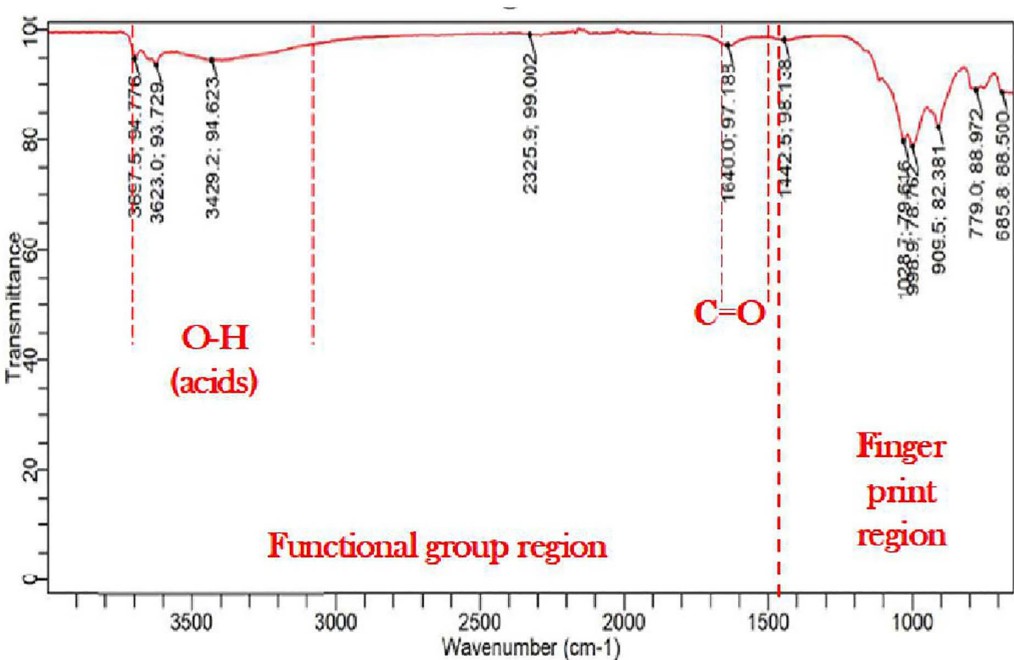

**Fig 14. FTIR results for the natural soil.**

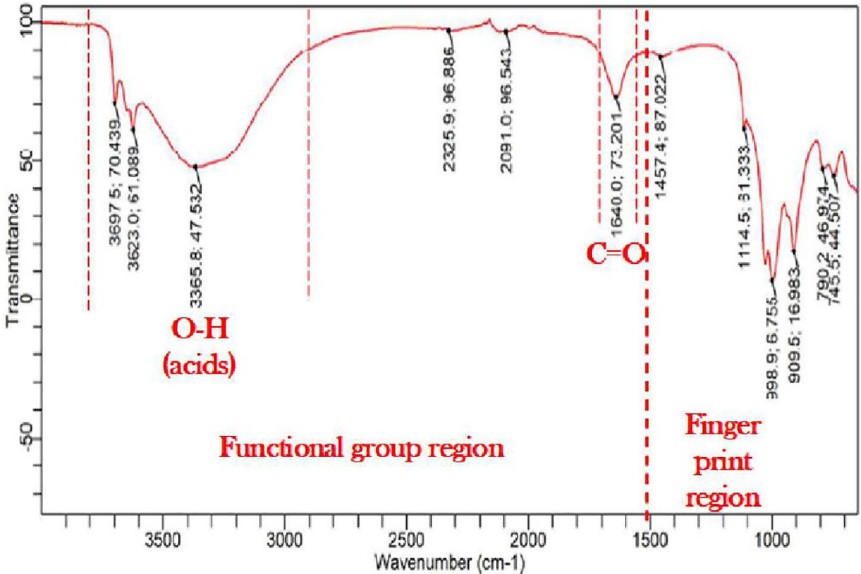

**Fig 15. FTIR results for the *Bacillus Anthracis* treated soil.**

i. The natural soil's CBR was improved by 80% at the 3 x $10^8$ cells ml$^{-1}$ treatment, moving from an S2 to an S3 class based on the Overseas Road Note 31, TRRL, and the Nigerian Federal Ministry of Works classification.

ii. The soil was improved from "very soft to soft" to "stiff to very stiff" for all *Bacillus Anthracis* treatments based on Obrzud and Truty's classification.

iii. The natural soil UCS value increased by 191% at 3.0 x $10^8$ cells ml$^{-1}$ *Bacillus Anthracis* treatment, confirming the CBR results.

iv. Best improvements at 3 x $10^8$ cells ml$^{-1}$ of *Bacillus Anthracis*-treated soil increased the time for the moisture to seep through the entire assumed 450 mm subgrade thickness to 28.54 and 37.39 days for BSL and BSH compactive effort, respectively, thereby providing better subgrade protection from water infiltration.

v. Calcite, which formed a link between the soil particles and filled the pore spaces, produced a more durable subgrade material, as explained by the decrease in volume change and weight loss after different wet and dry cycles. SEM images and FTIR results also confirmed the formation of calcite.

vi. Treated soils performed better in the field than in the laboratory.

The results successfully demonstrated the application of bio-improved laterite as a subgrade material for road construction. Though excellent improvement percentages on all tested soil properties were obtained, improved results were unsuitable for road construction in terms of strength due to the soil's poor initial CBR value. Hence, it is recommended that bio improvements be carried out on laterite soils with better CBR initial values during road construction.

## Acknowledgments

The authors would like to thank the Covenant University Centre for Innovation, Research, and Discovery (CUCRID) for supporting this work financially.

## Author contributions

**Conceptualization:** Reuben Sani, Tobi Vincent Ogunro, Isaac Ibukun Akinwumi.

**Data curation:** Reuben Sani.

**Formal analysis:** Reuben Sani.

**Funding acquisition:** Reuben Sani.

**Investigation:** Reuben Sani.

**Methodology:** Reuben Sani.

**Project administration:** Reuben Sani.

**Resources:** Reuben Sani.

**Software:** Reuben Sani.

**Supervision:** Tobi Vincent Ogunro, Isaac Ibukun Akinwumi.

**Validation:** Tobi Vincent Ogunro.

**Writing – original draft:** Reuben Sani.

**Writing – review & editing:** Tobi Vincent Ogunro, Isaac Ibukun Akinwumi.

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
