## [Decision Letter · Decision Letter 0]

19 Dec 2024

Dear Dr. Sani,

Thank you for submitting your manuscript to PLOS ONE. After careful consideration, we feel that it has merit but does not fully meet PLOS ONE’s publication criteria as it currently stands. Therefore, we invite you to submit a revised version of the manuscript that addresses the points raised during the review process.

We look forward to receiving your revised manuscript.

Kind regards,

Abhishek Sharma, PhD

Academic Editor

PLOS ONE

Journal Requirements:

3. We note that your Data Availability Statement is currently as follows: [All relevant data are within the manuscript and its Supporting Information files]

4. We note that Figures S2 and S3 includes an image of a participant in the study.

As per the PLOS ONE policy (http://journals.plos.org/plosone/s/submission-guidelines#loc-human-subjects-research ) on papers that include identifying, or potentially identifying, information, the individual(s) or parent(s)/guardian(s) must be informed of the terms of the PLOS open-access (CC-BY) license and provide specific permission for publication of these details under the terms of this license. Please download the Consent Form for Publication in a PLOS Journal (http://journals.plos.org/plosone/s/file?id=8ce6/plos-consent-form-english.pdf ). The signed consent form should not be submitted with the manuscript, but should be securely filed in the individual's case notes. Please amend the methods section and ethics statement of the manuscript to explicitly state that the patient/participant has provided consent for publication: “The individual in this manuscript has given written informed consent (as outlined in PLOS consent form) to publish these case details”.

5. Please upload a copy of Figures 1 to 15, to which you refer in your text on pages 9,10,11,12,13,14,15,16,18,19 and 20. If the figure is no longer to be included as part of the submission please remove all reference to it within the text.

Reviewers' comments:

Reviewer's Responses to Questions

**Comments to the Author**

1. Is the manuscript technically sound, and do the data support the conclusions?

Reviewer #1: Yes

Reviewer #2: Partly

2. Has the statistical analysis been performed appropriately and rigorously?

Reviewer #1: Yes

Reviewer #2: Yes

3. Have the authors made all data underlying the findings in their manuscript fully available?

Reviewer #1: Yes

Reviewer #2: Yes

4. Is the manuscript presented in an intelligible fashion and written in standard English?

Reviewer #1: Yes

Reviewer #2: Yes

Reviewer #1: This work aimed to investigate the effect of application of bio-improved laterite as a subgrade material for road construction using bacterium Bacillus Anthracis. The paper is written in understandable language and good structure, reflecting the high quality of the work. We have the following comments, which may be beneficial for the authors:

1. Section 2.3 shows a flowchart of the method of the work. I prefer to move this section to be the beginning of section 2 so the reader can see what steps the authors have taken and then explain each step in the following subsections.

2. Figure 7 require more explanation. what the numbers of the horizontal and vertical axis refer to?

How the author get the results shown in lines 310 to 312

3 . I could not find table 4

Regards,

Reviewer #2: The manuscript addresses an innovative and relevant topic: the application of microbial-induced calcite precipitation (MICP) for improving lateritic soils in road construction. The use of Bacillus Anthracis is novel, and the systematic experimental approach is commendable. However, the manuscript would benefit from clarifications, additional analysis, and refinement in several areas.

1. The rationale for selecting Bacillus Anthracis over other urease-positive bacteria is unclear. Include a justification, addressing potential environmental or safety implications of using this bacterium.

2. The introduction discusses the environmental benefits of MICP but lacks a comparative analysis with traditional soil stabilization methods like lime or cement. Adding this would strengthen the case for MICP.

3. The manuscript reports significant improvements in CBR and UCS values but does not clearly explain why the improvement plateaus at higher bacterial concentrations. This phenomenon should be discussed in detail.

4. While regression analysis results are included, confidence intervals and statistical significance for model coefficients are missing. Incorporating these would improve the credibility of the statistical approach.

5. The field testing section briefly mentions compaction methods but does not provide sufficient details on replication or error margins, which are necessary for validating field-to-lab result comparisons.

6. Key citations are missing in the manuscript. Please include references to support the claims and findings. Consider adding the following citations at suitable sections:

a. https://doi.org/10.1007/978-981-15-9554-7_10

b. https://doi.org/10.1007/978-981-16-6557-8_75

c. https://doi.org/10.1007/978-981-16-6557-8_78

d. https://doi.org/10.22044/jme.2023.13018.2361

e. https://doi.org/10.1080/15440478.2020.1848699

7. Figures presenting test results (e.g., load-penetration curves) lack consistent scales and annotations, making it difficult to interpret trends. Ensure uniform formatting and include detailed captions.

8. The study demonstrates an 80% improvement in CBR, but the final values remain below the threshold typically required for road subgrade materials. The conclusion should reflect this limitation more explicitly.

9. The methodology section does not explain why certain parameters (e.g., 3.0 × 10^8 cells/ml concentration) were chosen over others. A brief explanation or citation to prior studies would address this gap.

10. The study’s practical implications are discussed but lack a cost-benefit analysis comparing MICP treatment with traditional methods. Adding this analysis would provide a more comprehensive perspective.

11. The use of Bacillus Anthracis in MICP for road construction is novel, but the manuscript does not address potential scalability or challenges in deploying this approach at larger scales, which should be included in the discussion.

**Do you want your identity to be public for this peer review?** For information about this choice, including consent withdrawal, please see our Privacy Policy

Reviewer #1: No

Reviewer #2: **Yes: ** Dr. Akhilesh Nautiyal

---

## [Author Response · Author response to Decision Letter 1]

1 Mar 2025

Responses to all comments have been provided in the file named Response to Reviewers

---

## [Decision Letter · Decision Letter 1]

19 Jun 2025

Dear Dr. Sani,

Thank you for submitting your manuscript to PLOS ONE. After careful consideration, we feel that it has merit but does not fully meet PLOS ONE’s publication criteria as it currently stands. Therefore, we invite you to submit a revised version of the manuscript that addresses the points raised during the review process.

We look forward to receiving your revised manuscript.

Kind regards,

Vasanthavigar Murugesan, M.Sc., Ph.D.,https://orcid.org/my-orcid?orcid=0000

Academic Editor

PLOS ONE

**Additional Editor Comments:**

As per reviewers suggestion manuscript need major revision

Reviewers' comments:

Reviewer's Responses to Questions

**Comments to the Author**

Reviewer #1: (No Response)

Reviewer #2: All comments have been addressed

2. Is the manuscript technically sound, and do the data support the conclusions?

Reviewer #1: Partly

Reviewer #2: Partly

3. Has the statistical analysis been performed appropriately and rigorously?

Reviewer #1: N/A

Reviewer #2: Yes

4. Have the authors made all data underlying the findings in their manuscript fully available?

Reviewer #1: No

Reviewer #2: Yes

5. Is the manuscript presented in an intelligible fashion and written in standard English?

Reviewer #1: Yes

Reviewer #2: Yes

Reviewer #1: Dear Author,

I could not find the figures in the manuscript or in a separated file

Could you please attach them

Reviewer #2: The manuscript presents valuable insights; however, it would benefit significantly from a thorough English language revision to enhance clarity, coherence, and overall readability. Several sentences are grammatically incorrect or awkwardly phrased, which may hinder the reader's understanding of the content. The use of technical terms is sometimes inconsistent, and transitions between sections can be improved to ensure a smoother flow of ideas. Additionally, the manuscript contains typographical errors and improper punctuation in multiple places. It is recommended that the authors seek assistance from a professional English editing service or a native English-speaking colleague with academic writing experience. Improving the manuscript’s language quality will not only make the arguments more persuasive but also increase its impact and accessibility to a broader academic audience. Once the language issues are addressed, the manuscript's scientific merit and contributions will be more effectively communicated

**Do you want your identity to be public for this peer review?** For information about this choice, including consent withdrawal, please see our Privacy Policy

Reviewer #1: No

Reviewer #2: **Yes: ** AKHILESH NAUTIYAL

---

## [Author Response · Author response to Decision Letter 2]

23 Aug 2025

Reviewer 1,

Pace corrected Figures 1 - 15 where attached as a separate Zip file titled "Supporting Information - Compressed/ZIP File Archive" in accordance to PLOS ONE requirements, as supporting information in a Zip file which was might have not been forwarded along to you for review.

Reviewer 2:

A fully subscribed Grammarly package has been used to proofread the entire manuscript, addressing raised concerns

---

## [Decision Letter · Decision Letter 2]

7 Oct 2025

Experimental investigation of bio-improved laterite for use as a subgrade material in road construction

PONE-D-24-45961R2

Dear Dr. Sani,

We’re pleased to inform you that your manuscript has been judged scientifically suitable for publication and will be formally accepted for publication once it meets all outstanding technical requirements.

Kind regards,

Ahmed M. Yosri

Academic Editor

PLOS ONE

Additional Editor Comments (optional):

Reviewers' comments:

Reviewer's Responses to Questions

**Comments to the Author**

Reviewer #1: All comments have been addressed

2. Is the manuscript technically sound, and do the data support the conclusions?

Reviewer #1: Yes

3. Has the statistical analysis been performed appropriately and rigorously?

Reviewer #1: Yes

4. Have the authors made all data underlying the findings in their manuscript fully available?

Reviewer #1: Yes

5. Is the manuscript presented in an intelligible fashion and written in standard English?

Reviewer #1: Yes

Reviewer #1: (No Response)

**Do you want your identity to be public for this peer review?** For information about this choice, including consent withdrawal, please see our Privacy Policy

Reviewer #1: No

---

## [Editor Report · Acceptance letter]

PONE-D-24-45961R2

PLOS ONE

Dear Dr. Sani,

I'm pleased to inform you that your manuscript has been deemed suitable for publication in PLOS ONE. Congratulations! Your manuscript is now being handed over to our production team.

Kind regards,

on behalf of

Dr. Ahmed M. Yosri

Academic Editor

PLOS ONE